# Plasmid-mediated antibiotic resistance among uropathogens in primigravid women —Hyderabad, India

Nagamani Kammili[1]*, Manisha Rani[1], Ashley Styczynski[2], Madhavi latha[3], Panduranga Rao Pavuluri[4], Vishnuvardhan Reddy[4], Marcella Alsan[2]

1 Department of Microbiology, Gandhi Medical College and Hospital, Secunderabad, Telangana, India,
2 Division of Infectious Diseases and Geographic Medicine, Department of Medicine, Stanford University, Palo Alto, California, United States of America, 3 MDRU, Gandhi Medical College, Secunderabad, Telangana, India, 4 Ella Foundation, Bharathbiotech, Hyderabad, Telangana, India

* nagamaniy2k03@rediffmail.com

**Data Availability Statement:** All relevant data are within the paper and its Supporting Information files.

**Funding:** Freeman Spogli Institute for International Studies, Stanford University http://dx.doi.org/10.

## Abstract

With the growing threat of antimicrobial resistance worldwide, uncovering the molecular epidemiology is critical for understanding what is driving this crisis. We aimed to evaluate the prevalence of plasmid-mediated-quinolone-resistance (PMQR) and extended-spectrum beta-lactamase- (ESBL) producing gram-negative organisms among primigravid women with bacteriuria. We collected urine specimens from primigravid women attending their first antenatal visit at Gandhi Hospital during October 1, 2015 to September 30, 2016. We determined antimicrobial susceptibility and ESBL and quinolone resistance using VITEK-2. We performed polymerase chain reaction amplification on resistant isolates for detection of ESBL-encoding genes (*TEM*, *SHV*, *CTX-M*) and PMQR genes (*qnrA*, *qnrB*, *qnrD*, *qnrS*, *aac (6')-Ib-cr*). Of 1,841 urine samples, 133 demonstrated significant bacterial growth with gram-negative bacilli accounting for 85% of isolates, including *Escherichia coli* (n = 79), *Klebsiella pneumoniae* (n = 29), *Sphingomonas* (n = 3), *Enterobacter* (n = 1), and *Citrobacter* (n = 1). We found 65% of *E. coli* isolates and 41% of *K. pneumoniae* isolates were ESBL positive. Of ESBL-positive isolates, the most common genes conferring resistance were *TEM-1* (66.7%) followed by *CTX-M-15* (33.3%). Fifty-seven percent of ESBL-positive *E. coli* also demonstrated resistance to quinolones with the most common PMQR genes being *qnr-S* (62.5%) and *aac (6')-Ib-cr* (37.5%). We did not find any resistance to quinolones among ESBL-positive *K. pneumoniae* isolates. Across different classes of antibiotics we found a strong clustering of multi-drug resistance in *E. coli* with over 45% of ESBL-positive isolates demonstrating resistance to at least three classes of antibiotics. This study emphasizes the high prevalence of plasmid-mediated ESBL and quinolone resistance in community-acquired urinary tract infections of primigravid women. The overall abundance of multi-drug-resistant isolates in this population is alarming and may present therapeutic challenges.

13039/100006100 Dr.Manisha Rani one of the co-author was recruited as Research Assistant in this project and received salary. The funders had no role in study design, data collection and analysis, decision to publish, or preparation of the manuscript.

**Competing interests:** The authors have declared that no competing interests exist.

## Introduction

Emergence of community-acquired multi-drug-resistant bacterial infections poses a grave public health threat. Urinary tract infections (UTIs) are a major proportion of community-acquired infections that have demonstrated increasing patterns of antimicrobial resistance. UTIs occur in 2–10% of pregnant women, which may be symptomatic or asymptomatic [1]. Regardless of symptoms, untreated or undertreated bacteriuria in pregnancy increases risk for adverse outcomes including preterm birth, low birth weight, and pyelonephritis, which can lead to excess maternal and neonatal morbidity and mortality [2–4]. Thus, screening and treating pregnant women for bacteriuria has become a routine part of prenatal care [5, 6]. Evaluating the bacteriological profiles of bacteriuria in pregnant women attending antenatal clinics provides an opportunity to study the prevalence of antimicrobial resistance in community-acquired uropathogens and determine appropriateness of empiric treatment options.

Cephalosporins and combination beta-lactam/beta-lactamase inhibitors are considered first-line therapy in the treatment of bacteriuria in pregnancy. Similarly, cephalosporins and fluoroquinolones are frequently used for treating community-acquired UTIs in non-pregnant adults due to their potency, broad spectrum of activity, oral bioavailability, and safety profile [7]. However, with increasing antibiotic resistance worldwide, the efficacy of these antibiotic treatment options may be threatened.

Extended-spectrum beta-lactamases (ESBLs) are a group of genetic mutations that confer resistance by hydrolysing penicillins, first-, second-, and third-generation cephalosporins, and aztreonam. They can be inhibited by beta-lactamase inhibitors. ESBLs are encoded by three major groups of genes: *TEM*, *SHV*, and *CTX-M* [8], and these enzymes are often found in *Escherichia coli* and *Klebsiella pneumoniae* [9]. Several different species of bacteria are capable of producing ESBLs, which were initially associated with healthcare-associated infections (HCAIs), but are increasingly being associated with community-acquired infections.

Fluoroquinolones are used to treat UTIs caused by both gram-positive and gram-negative bacteria. Wide usage of these antibiotics has led to resistance, especially among Enterobacteriaceae [10]. Fluoroquinolone resistance varies from 2.2% to 69% among community-acquired UTIs [11]. The emergence of plasmid-mediated quinolone resistance (PMQR) was first found in a strain of *K. pneumonia* in the USA in 1998 and shown to be due to a member of the penta-peptide repeat family of proteins qnr [12]. Qnr interacts with DNA gyrase and topoisomerase IV to prevent quinolone inhibition. In subsequent years, several distantly-related plasmid-mediated qnr determinants have been described in Enterobacteriaceae (*qnrB, qnrC, qnrD, qnrS*). The qnr genes are usually integrated into plasmids, which make them particularly susceptible to cross-species transmission [13]. In addition, these plasmids often harbor other antibiotic resistance genes such as ESBLs and favor the selection and dissemination of fluoroquinolone-resistant strains by chemically unrelated drug classes and vice versa [14].

There is limited information regarding the frequency of ESBL and fluoroquinolone-resistance genes in community-acquired infections in India. Therefore, this study aimed to uncover the epidemiology of PMQR and ESBL genes in gram-negative bacilli among primigravid women with bacteriuria attending the antenatal clinic for the first time.

## Materials and methods

### Study design and sampling

A cross-sectional observational study was conducted at Gandhi Medical College and Hospital, Hyderabad from October 1, 2015 to September 30, 2016. We collected urine specimens from primigravid women attending their first antenatal clinic visit at the outpatient department of

Gandhi Hospital. We excluded multigravid and primigravid women who previously attended antenatal clinics to rule out the possibility of any HCAIs or colonization resulting from health-care exposure. In addition, we administered surveys regarding community exposures at the time of enrollment that indicated recent antibiotic use was rare among study participants [15]. The study was approved by IRB committees from Gandhi Medical College and Hospital, Stanford University, and the Indian Council of Medical Research. We also obtained written informed consent from all study participants before specimen collection and interviews.

A structured case proforma was filled by a trained investigator during the antenatal visit. The questionnaire was based on the Demographic and Health Surveys tool AMR Module for Population-Based Surveys [16]. The questionnaire included their usual and current residence, occupation, husband's occupation, household income, religion, caste, education level, dietary and hygiene practices, and recent non vitamin tablet consumption. As most of the women were unaware of the antimicrobial drug use, ingestion of any other tablet other than vitamin was the question asked for antimicrobial drug use.

## Antimicrobial susceptibility testing

We performed phenotypic antimicrobial susceptibility testing and ESBL and quinolone resistance screening using a VITEK-2 system in accordance with CLSI guidelines [17].

## Genotypic characterization of ESBL and PMQR genes

We targeted ESBL-producing and fluoroquinolone-resistant isolates for detection of ESBL-encoding genes (*TEM*, *SHV*, *CTX-M*) and PMQR-encoding genes (*qnrA*, *qnrB*, *qnrD*, *qnrS*, *aac (6')-Ib-cr*), respectively. We isolated DNA from bacterial cells by using a boiling method. For polymerase chain reaction (PCR) amplification, we prepared master mix using 10 μl dNTPs, 2μl of each primer, 1 μl of Taq polymerase in 6.5 μl PCR buffer, and 22 μl of RNase-free water. We added DNA to the above mixture to a final volume of 50 μl [18], utilizing published primers (Tables 1 and 2) [19, 20]. We performed amplification in a thermocycler according to the following cycling parameters: initial denaturation step at 95˚C for 5minutes, 35 cycles of denaturation at 95˚C for 30 seconds, annealing at 60˚C for 30 seconds, extension at 72˚C for 2 minutes, final extension step at 72˚C for 10 minutes, and a hold at 4˚C. We subjected amplified PCR products to agarose gel electrophoresis with 1.8% agarose gel using suitable molecular weight markers. We visualized the gel on a UV platform in a gel documentation system (Figs 1 and 2).

**DNA sequencing.** We sequenced the purified PCR products with an ABI 3730XL sequencer (Applied Biosystems, USA). We analysed the nucleic acid sequences using the Basic Local Alignment Search Tool available at the National Centre for Biotechnology database. We submitted the nucleic acid sequences to Genbank (accession numbers for ESBLs: MH745708, MH745709, MH745710, MH745711, MH745712, MH745713, MH745714, MH745715, MH745716 and PMQRs: MK761221, MK761222, MK761223, MK761224, MK761225, and MK761226).

## Results

### Distribution of ESBL and quinolone resistance among isolates

Of 1,841 urine samples, 133 had significant bacterial growth, defined as $\geq 10^5$ CFU/mL. Gram-negative bacilli accounted for 85% (113) of the isolates, including *E. coli* (n = 79), *K. pneumoniae* (n = 29), *Sphingomonas* (n = 3), *Enterobacter* (n = 1), and *Citrobacter* (n = 1) (Fig 3).

**Table 1. Primers for polymerase chain reaction of ESBL genes.**

| Primer | Orientation | Oligonucleotide sequence (5-3) | Size (bp) |
|--------|-------------|-------------------------------|-----------|
| CTX-M | Forward | 5'- GAAGGTCATCAAGAAGGTGCG -3' | 560bp |
| | Reverse | 5'- GCATTGCCACGCTTTTCATAG- 3' | |
| TEM | Forward | 5'—GAGACAATAACCCTGGTAAAT- 3' | 459 bp |
| | Reverse | 5'- AGAAGTAAGTTGGCAGCAGTG- 3' | |
| SHV | Forward | 5'-GTCAGCGAAAAACACCTTGCC- 3' | 383bp |
| | Reverse | 5'-GTCTTATCGGCGATAAACCAG- 3' | |

Based on VITEK-2 determinations, we detected ESBL positivity in 65% (51) of *E. coli* isolates and 41% (12) of *K. pneumonia* isolates. Quinolone resistance was observed in 47% (37) of *E. coli* isolates, whereas only one isolate of *K. pneumoniae* demonstrated resistance to quinolones.

## Resistance patterns to other antibiotics

We evaluated for resistance against individual antimicrobial agents separated by ESBL determination. Among ESBL-positive *E.coli* isolates, we observed the highest resistance against nalidixic acid (86%), which can signify reduced susceptibility to fluoroquinolones. High levels of resistance were also noted for ciprofloxacin (57%), trimethoprim/sulfamethoxazole (55%), and gentamicin (33%). Multi-drug resistance (resistance to at least 3 classes of antibiotics) was noted in 45% of ESBL-positive *E. coli* isolates (Table 3).

Among ESBL-positive *K. pneumoniae* isolates, rates of resistance to other antibiotics was lower, though a substantial number of isolates demonstrated only intermediate susceptibility to nitrofurantoin (50%), a common treatment option for community-acquired UTIs. Resistance to other antimicrobial classes are shown in Table 4. All the ESBL-positive *K. pneumoniae* isolates were sensitive to nalidixic acid and ciprofloxacin, and only one isolate demonstrated multi-drug resistance (8%).

## Distribution of ESBL genes

We identified 63 phenotypically-confirmed ESBLs (51 *E. coli* and 12 *K. pneumoniae)*, which were genotypically characterized for ESBL genes (*CTX-M*, *SHV*, *and TEM*). Among these isolates, the most common ESBL gene was *TEM-1* in *E. coli* (62.7%) and *K. pneumoniae* (83.3%), followed by *CTX-M-15* as the second most prevalent gene at 35.2% and 25%, respectively

**Table 2. Primers for polymerase chain reaction of PMQR genes.**

| Primer | Orientation | Oligonucleotide sequence (5-3) | Size (bp) | Annealing Temperature |
|--------|-------------|-------------------------------|-----------|----------------------|
| qnrA | Forward | CAGCAAGAGGATTTCTCACG | 630 bp | 58˚C |
| | Reverse | AATCCGGCAGCACTATTACTC | | |
| qnrB | Forward | GGCTGTCAGTTCTATGATCG | 488 bp | 59.1˚C |
| | Reverse | SAKCAACGATGCCTGGTAG | | |
| qnrD | Forward | CGAGATCAATTTACGGGGAATA | 581 bp | 57˚C |
| | Reverse | AACAAGCTGAAGCGCCTG | | |
| qnrS | Forward | GCAAGTTCATTGAACAGGGT | 428 bp | 55.6˚C |
| | Reverse | TCTAAACCGTCGAGTTCGGCG | | |
| aac (6')-Ib-cr | Forward | TTGGAAGCGGGGACGGAM | 260 bp | 58˚C |
| | Reverse | ACACGGCTGGACCATA | | |

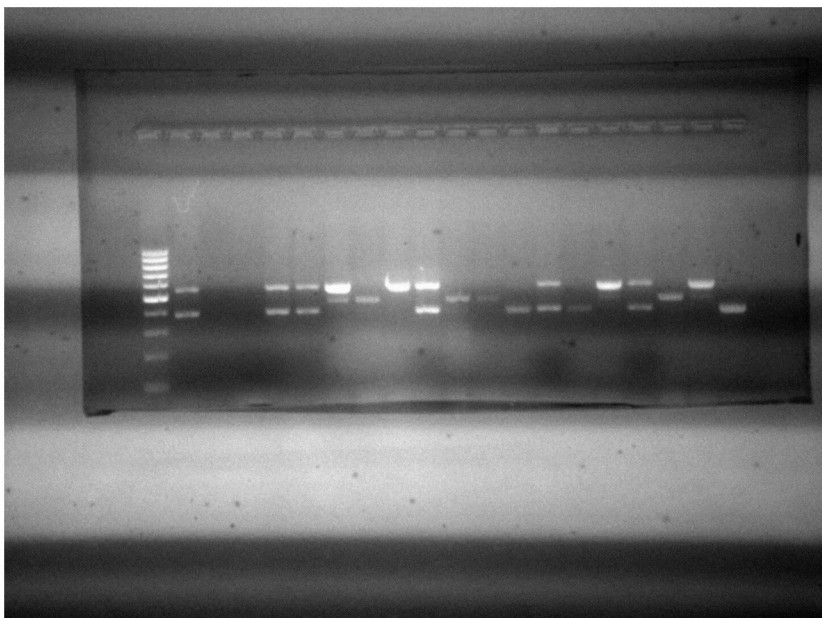

**Fig 1. Gel electrophoresis detection of ESBL-producing genes among *E. coli* and *K. pneumoniae* isolates.** Results by lane: 1- ladder (100bp); 2- *K. pneumoniae* (ATCC 700603) *CTX-M-15+SHV-38* genes; 3-*E. coli* (ATCC 25922); 4-Undetected; 5,6,10,14,17- *CTX-M-15+SHV-38* genes; 13,15,20- *SHV-38* gene; 7,16,19- *CTX-M-15 +TEM-1* genes; 8,11,12,18- *TEM-1* gene.

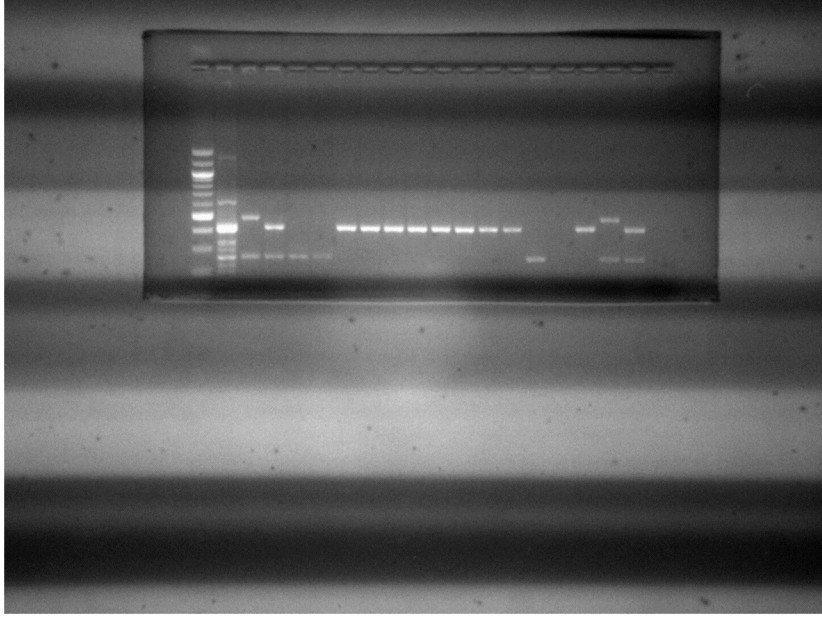

**Fig 2. Gel electrophoresis detection of PMQR genes among *E. coli* and *K. pneumoniae* isolates.** Results by lane: 1-ladder (100bp); 2- ladder (50bp); 3- *K. pneumonia qnrB+aac (6')-lb-cr* genes; 4- *qnrS+aac (6')-lb-cr* genes; 5,6,15- *aac (6')-lb-cr* genes; 7–14- *qnrS* gene; 16-Undetected.

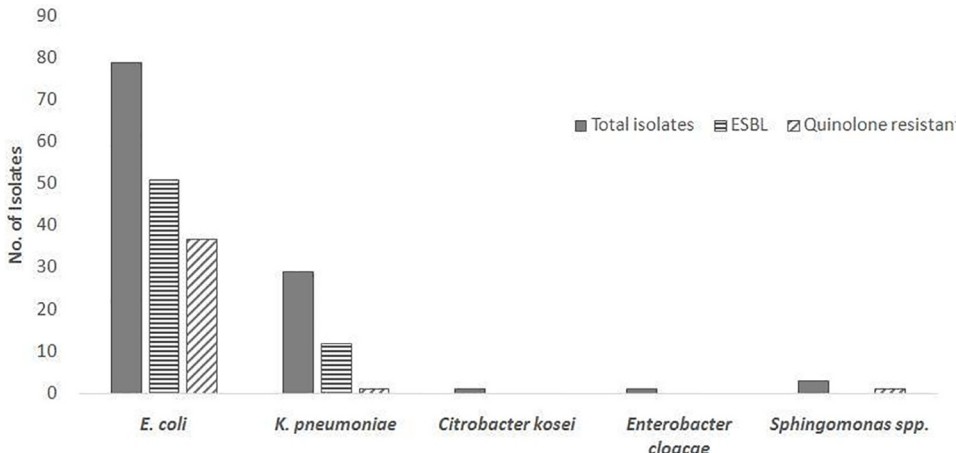

**Fig 3. Phenotypic distribution of ESBL and quinolone resistance among isolates.** High levels of ESBL and quinolone resistance were observed among *E. coli* isolates. *K. pneumoniae* isolates demonstrated less but still substantial resistance.

(Table 5). We found a co-occurrence of *CTX-M-15* and *TEM-1* genes in 23.8% of isolates and *TEM-1* and *SHV-38* genes in 4.8%. Overall 28.6% of isolates carried two resistance genes.

## Distribution of PMQR genes

We identified 43 phenotypically-confirmed quinolone-resistant ESBL *E. coli* strains, and we genotypically characterized 32 for PMQR genes. The most common gene in the present study was *qnrS* at 62.5% followed by *aac (6′)-Ib-cr* at 37.5% (Table 6). *QnrA*, *qnrB*, and *qnrD* were not detected. Since all of the ESBL-positive *K. pneumoniae* were sensitive to quinolones they were not tested for PMQR genes.

## Discussion

The aim of the study was to evaluate the prevalence of PMQR and ESBL-producing gram-negative organisms among primigravid women with bacteriuria. Pregnant women are an

**Table 3. Antimicrobial resistance patterns of *E. coli* isolates by ESBL positivity.**

| Antibiotic | ESBL *E. coli* (n = 51) | | | Non-ESBL *E. coli* (n = 28) | | |
|---|---|---|---|---|---|---|
| | S | I | R | S | I | R |
| Amoxicillin/Clavulanic acid | 28 (56%) | 18 (36%) | 4 (8%) | 26 (92.9%) | 2 (7.1%) | - |
| Cefuroxime | 10 (20%) | 1 (2%) | 39 (78%) | 25 (89.3%) | 1 (3.6%) | 2 (7.1%) |
| Ceftriaxone | 13 (26%) | 2 (4%) | 35 (70%) | 24 (88.9%) | 1 (3.7%) | 2 (7.4%) |
| Piperacillin/Tazobactam | 43 (86%) | 3 (6%) | 4 (8%) | 25 (96.2%) | - | 1 (3.8%) |
| Gentamicin | 34 (66.7%) | - | 17 (33.3%) | 28 (100%) | - | - |
| Nalidixic acid | 7 (14%) | - | 43 (86%) | 11 (39.3%) | - | 17 (60.7%) |
| Ciprofloxacin | 21 (41.2%) | 1 (2.0%) | 29 (56.9%) | 20 (71.4%) | 1 (3.6%) | 7 (25%) |
| Meropenem | 51 (100%) | - | - | 28 (100%) | - | - |
| Nitrofurantoin | 48 (96%) | 1 (2.0%) | 1 (2.0%) | 25 (89.3%) | 2 (7.1%) | 1 (3.6%) |
| Trimethoprim/sulfamethoxazole | 23 (45.1%) | - | 28 (54.9%) | 20 (76.9%) | - | 6 (23.1%) |
| Multi-drug resistant (≥3 classes) | | | 23 (45.1%) | | | 0 |

**Table 4. Antimicrobial resistance patterns of *K. pneumoniae* isolates by ESBL positivity.**

| Antibiotic | ESBL *K. pneumoniae* (n = 12) | | | Non-ESBL *K. pneumoniae* (n = 17) | | |
|---|---|---|---|---|---|---|
| | S | I | R | S | I | R |
| Amoxicillin/Clavulanic acid | 11 (91.7%) | - | 1 (8.3%) | 17 (100%) | - | - |
| Cefuroxime | 3 (25%) | - | 9 (75%) | 17 (100%) | - | - |
| Ceftriaxone | 2 (16.7%) | - | 10 (83.3%) | 17 (100%) | - | - |
| Piperacillin/Tazobactam | 12 (100%) | - | - | 17 (100%) | - | - |
| Gentamicin | 11 (91.7%) | - | 1 (8.3%) | 17 (100%) | - | - |
| Nalidixic acid | 12 (100%) | - | - | 15 (88.2%) | - | 2 (11.8%) |
| Ciprofloxacin | 12 (100%) | - | - | 16 (94.1%) | - | 1 (5.9%) |
| Meropenem | 12 (100%) | - | - | 17 (100%) | - | - |
| Nitrofurantoin | 5 (41.7%) | 6 (50.0%) | 1 (8.3%) | 12 (70.6%) | 5 (29.4%) | - |
| Trimethoprim/sulfamethoxazole | 12 (100%) | - | - | 17 (100%) | - | - |
| Multi-drug resistant (≥3 classes) | | | 1 (8.3%) | | | 0 |

understudied population with regards to antimicrobial resistance, and very few studies have previously examined the prevalence of PMQR and ESBL genes among enterobacterial isolates from pregnant women [21]. However, the implications of the findings likely extend to uropathogen resistance profiles in the community given the absence of prior healthcare exposure in this population. The high overall rates of plasmid-mediated resistance in this study are cause for concern.

Since the early 2000s a change in epidemiology of ESBL-producing Enterobacteriaceae was observed with increasing reports of their occurrence in community-acquired infections [22]. We detected an overall prevalence of ESBL-producing *E. coli* isolates of 65% in this study with 48% phenotypic resistance to third-generation cephalosporins. This contrasts with studies published from 2011–2014 evaluating bacteriuria in pregnant women in India that found rates of *E. coli* resistance to cephalosporins as low as 0–14% [23, 24]. Similarly, a previous study of rectal *E. coli* isolates from pregnant women in India demonstrated 17–19% resistance to third-generation cephalosporins, with 86% of those isolates producing ESBLs, highlighting either the rapid spread of antibiotic resistance or substantial variations in local epidemiology [21].

The emergence of ESBL-producing Enterobacteriaceae was attributed to the spread of the *CTX-M* gene [22]. However, the most common ESBL gene in the present study was *TEM-1*, similar to findings from other studies in India [25–27]. Additionally, in our study a third of isolates carried more than one type of beta-lactamase gene, similar to prior reports that have demonstrated a frequent co-occurrence of ESBL genes [22, 27]. Overall, 84% percent of ceftriaxone-resistant isolates carried at least one ESBL gene. Several studies in India have shown *Klebsiella spp.* as the major ESBL producer [25, 26], though our study revealed greater ESBL positivity among *E. coli*, which has also been reflected in other studies [28].

**Table 5. Frequency of ESBL genes among ESBL-producing *E. coli* and *K. pneumoniae*.**

| ESBL Gene | *E. coli* (n = 51) | *K. pneumoniae* (n = 12) |
|---|---|---|
| *CTX-M-15* | 18 (35.2%) | 3 (25.0%) |
| *TEM-1* | 32 (62.7%) | 10 (83.3%) |
| *SHV-38* | 10 (19.6%) | 2 (16.6%) |
| *TEM-1+SHV-38* | 3 (5.8%) | 0 |
| *CTX-M-15+TEM-1* | 12 (23.5%) | 3 (25.0%) |

**Table 6. Frequency of PMQR genes among quinolone-resistant, ESBL-positive *E. coli*.**

| PMQR Gene | *E. coli* (n = 32) | Nalidixic acid resistance | Ciprofloxacin Resistance |
|---|---|---|---|
| *qnrA* | 0 (0%) | 0 | 0 |
| *qnrB* | 0 (0%) | 0 | 0 |
| *qnrD* | 0 (0%) | 0 | 0 |
| *qnrS* | 20 (62.5%) | 20 | 15 |
| *aac (6')-Ib-cr* | 12 (37.5%) | 9 | 9 |
| *qnrS +aac (6')-Ib-cr* | 2 (6.25%) | 9 | 4 |

We demonstrated a high level of quinolone resistance among ESBL-positive *E. coli*, supporting the co-transmission of resistance genes on plasmids. This has been seen in other studies in India where 61% of ciprofloxacin-resistant *E. coli* isolates demonstrated ESBL production [29]. Accordingly, multi-drug resistance was common in *E. coli*, though lower resistance was seen in *K. pneumoniae* isolates. As *E. coli* is one of the most common causes of bacteriuria in pregnant women as well as community-acquired UTIs, the clustering of resistance genes among *E. coli* isolates presents concerning therapeutic challenges.

Fluoroquinolone therapy is generally avoided in pregnant women because of safety concerns in developing foetuses, though it remains an important treatment for UTIs and other types of infections in non-pregnant individuals. Overall resistance to fluoroquinolones in this study both in ESBL and non-ESBL varied substantially between *E. coli* (45.1%) and *K. pneumonia* (3.6%). Qnr genes were detected in 86% of quinolone-resistant isolates. This is higher than another study from India that reported a prevalence of 50% among ciprofloxacin-resistant uropathogenic *E. coli* [29]. The total percentage of *E. coli* isolates harbouring qnr genes in this study, as shown in Table 3, is also significantly higher compared to Brazil, Europe, the USA, and elsewhere in Asia [19, 30–35].

In this study, we only detected *qnrS* and *aac (6')-Ib-cr* genes in *E. coli*, with almost a quarter of isolates containing both genes. *QnrA*, *qnrB*, and *qnrC* were not found. Few studies in India reported PMQR genes among clinical isolates of Enterobacteriaceae, but the most prevalent gene reported has been *aac (6')-lb-cr* [21, 29, 36]. However, regionally *qnrS* has been found to be a major antimicrobial resistant gene in environmental samples in India [36]. Studies on *E. coli* strains showing higher prevalence of *qnrA* and *qnrB* have mostly been among clinical isolates from HCAIs [31].

The increased frequency of ESBL and PMQR genes detected in this study carries significant implications for the management of community-acquired infections and further spread of these plasmid-mediated resistance mechanisms through horizontal gene transfer to other bacterial classes. Additionally, it raises the question about what is driving this trend. Overuse and misuse of antibiotics has changed the landscape of resistance, but this study population has not been directly influenced by the typical risk factors such as recent antibiotic use or healthcare contact/hospitalization [21, 37, 38]. After eliminating the effects of healthcare, presence of resistant bacteria has been found to be most closely linked to low socioeconomic status [15]. In fact, a review of community-acquired UTIs found that exposure to food, animal, and environmental sources may predict UTIs caused by ESBL-producing *E. coli* [39]. This finding suggests that there are environmental selective pressures that are distinct from the healthcare setting, and contamination of the environment by biocides and antibiotic residues are increasing resistance in community flora [40–42].

Indeed, in a series of studies conducted near the present research site, high levels of fluoroquinolones have been detected in water and environmental sources as a result of

contamination by pharmaceutical manufacturing effluent [43, 44]. Environmental contamination in particular selects for antibiotic resistance genes on mobile genetic elements because it promotes increased uptake of foreign DNA, which contributes to bacterial resilience under selective pressures [45, 46]. The findings of this study provide concerning evidence that these environmental processes may be spreading into human pathogens.

In conclusion, our study emphasizes the high prevalence of plasmid-mediated ESBL and quinolone resistance in community-acquired UTIs. While lower resistance was found in *K. pneumoniae* isolates compared with *E. coli*, overall abundance of drug resistance among uropathogens in this population is alarming. The findings of this study limit empiric treatment options for community-acquired UTI, including among pregnant women. Early detection of multi-drug-resistant isolates in routine microbiology laboratories is critical to avoid treatment failure and the complications thereof. Future surveillance and characterization of plasmids carrying multi-drug-resistance determinants could improve understanding of the origin and evolution of gram-negative bacterial resistance and inform infection control efforts.

## Supporting information

**S1 Data.**
(XLSX)

## Author Contributions

**Conceptualization:** Nagamani Kammili, Manisha Rani, Ashley Styczynski, Vishnuvardhan Reddy, Marcella Alsan.

**Data curation:** Nagamani Kammili, Manisha Rani, Marcella Alsan.

**Formal analysis:** Nagamani Kammili, Manisha Rani, Panduranga Rao Pavuluri, Vishnuvardhan Reddy, Marcella Alsan.

**Funding acquisition:** Marcella Alsan.

**Investigation:** Manisha Rani, Ashley Styczynski, Marcella Alsan.

**Methodology:** Nagamani Kammili, Manisha Rani, Ashley Styczynski, Marcella Alsan.

**Project administration:** Nagamani Kammili, Manisha Rani, Marcella Alsan.

**Resources:** Madhavi latha, Marcella Alsan.

**Software:** Marcella Alsan.

**Supervision:** Nagamani Kammili, Ashley Styczynski, Panduranga Rao Pavuluri, Vishnuvardhan Reddy, Marcella Alsan.

**Validation:** Manisha Rani, Ashley Styczynski, Marcella Alsan.

**Visualization:** Marcella Alsan.

**Writing – original draft:** Nagamani Kammili, Manisha Rani, Ashley Styczynski, Madhavi latha, Marcella Alsan.

**Writing – review & editing:** Nagamani Kammili, Ashley Styczynski, Marcella Alsan.

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
