## [Decision Letter · Decision Letter 0]

10 Dec 2019

PONE-D-19-29203

High levels of plasmid-mediated antibiotic resistance among uropathogens in primigravid women – Hyderabad, India

PLOS ONE

Dear Dr Kammili,

Thank you for submitting your manuscript to PLOS ONE. After careful consideration, we feel that it has merit but does not fully meet PLOS ONE’s publication criteria as it currently stands. Therefore, we invite you to submit a revised version of the manuscript that addresses the points raised during the review process.

Two reviewers have commented on the manuscript. They have pointed out the need for clarifications in several sections of the manuscript. The manuscript needs revision addressing the referee comments

We would appreciate receiving your revised manuscript by Jan 24 2020 11:59PM. To enhance the reproducibility of your results, we recommend that if applicable you deposit your laboratory protocols in protocols.io, where a protocol can be assigned its own identifier (DOI) such that it can be cited independently in the future. For instructions see: http://journals.plos.org/plosone/s/submission-guidelines#loc-laboratory-protocols

We look forward to receiving your revised manuscript.

Kind regards,

Iddya Karunasagar

Academic Editor

PLOS ONE

Additional Editor Comments:

Two reviewers have commented on the manuscript. They have pointed out the need for clarifications in several sections of the manuscript. The manuscript needs revision addressing the referee comments.

2. In your Methods section, please provide additional information about the participant recruitment method and the demographic details of your participants.

Please ensure you have provided sufficient details to replicate the analyses such as:

a) the recruitment date range (month and year), 

b) a description of how participants were recruited, and

c) descriptions of where participants were recruited and where the research took place.

'The funders had no role in study design, data collection and analysis, decision to publish, or preparation of the manuscript.'

Please provide an amended Funding Statement that declares *all* the funding or sources of support received during this specific study (whether external or internal to your organization) as detailed online in our guide for authors at http://journals.plos.org/plosone/s/submit-now Please state what role the funders took in the study.  If any authors received a salary from any of your funders, please state which authors and which funder. If the funders had no role, please state: "The funders had no role in study design, data collection and analysis, decision to publish, or preparation of the manuscript."

6. Please upload a copy of Supporting Information Table S1 which you refer to in your text on page 21.

7. Please include captions for your Supporting Information files at the end of your manuscript, and update any in-text citations to match accordingly. Please see our Supporting Information guidelines for more information: http://journals.plos.org/plosone/s/supporting-information

Reviewers' comments:

Reviewer's Responses to Questions

**Comments to the Author**

1. Is the manuscript technically sound, and do the data support the conclusions?

Reviewer #1: Yes

Reviewer #2: Partly

2. Has the statistical analysis been performed appropriately and rigorously? 

Reviewer #1: No

Reviewer #2: No

3. Have the authors made all data underlying the findings in their manuscript fully available?

Reviewer #1: Yes

Reviewer #2: No

4. Is the manuscript presented in an intelligible fashion and written in standard English?

Reviewer #1: Yes

Reviewer #2: Yes

5. Review Comments to the Author

Reviewer #1: It is mentioned that consent was taken for interview.(In addition, we administered surveys regarding community exposures at the time of enrollment) But there is no discussion on outcome of the interviews?

Appropriate statistical analysis has not been made to highlight the proportion of various genes for ESBLs in the isolates.

Sequencing outcomes have not been discussed.Was any mutation detected?

What is the explanation for isolates with undetected genes? No explanation has not been given.

Limitations of the study have not been discussed.

Reviewer #2: as enclosed as attachment.

• In the methodology its mentioned survey. What kind of surveys were carried out among primigravid and what information was collected details needs to be included.

• During the discussion it has been mentioned that molecular epidemiology ESBL and quinolone producing gram negative drug resistant isolates in primigravid during first visit of primigravid antenatal clinic has been described in this manuscript. However that has not been the aim of the study.

• In conclusion also study emphasises on high prevalence of drug resistant to ESBL and quinolone in community acquired UTI. Which is not included in the aims.

• In the study there is no molecular epidemiology data provided so as to corroborate with discussion as mentioned on the line no 201 to 204.

• More over its only drug resistance study using different methods including molecular levels and gene detection.

• As the study has not been compared with any other groups or with previous year data title of the study is deceptive. Needs to be corrected.

Clarification on above points may be provided

6. PLOS authors have the option to publish the peer review history of their article (what does this mean?). If published, this will include your full peer review and any attached files.

Reviewer #1: Yes: Reba Kanungo

Reviewer #2: No

---

## [Author Response · Author response to Decision Letter 0]

7 Feb 2020

Comments on plasmid-mediated antibiotic resistance among uropathogens in primigravid women – Hyderabad, India: 

The above study was carried out with the aim of to evaluate the prevalence of plasmid-mediated-quinolone-resistance (PMQR) and extended-spectrum beta lactamase- (ESBL) producing gram-negative organisms among primigravid women with bacteriuria. This has been done to uncover the epidemiology of PMQR and ESBL 84 genes in gram-negative bacilli among primigravid women having urinary infections. Comments are given below: 

• In the methodology its mentioned survey. What kind of surveys were carried out among primigravid and what information was collected details needs to be included.

Response to Reviewer: 

A cross- sectional observational study was conducted at Gandhi Medical College and Hospital. The information collected details were included in the previous published paper. Reference number 15 in the present article. 

Alsan M, Kammili N, Lakshmi J, Xing A, Khan A, Rani M, et al. Poverty and Community-Acquired Antimicrobial Resistance with Extended-Spectrum β-Lactamase–Producing Organisms, Hyderabad, India. EmergInfect Dis. 2018 Aug; 24(8):1490.

• During the discussion it has been mentioned that molecular epidemiology ESBL and quinolone producing gram negative drug resistant isolates in primigravid during first visit of primigravid antenatal clinic has been described in this manuscript. However that has not been the aim of the study.

Response to Reviewer: 

Correction has been made and included in page 4 and 5, lines 85,86,87. 

• In conclusion also study emphasises on high prevalence of drug resistant to ESBL and quinolone in community acquired UTI. Which is not included in the aims. 

Response to Reviewer: 

It has been included in page 4 and 5, lines 85,86,87.

• In the study there is no molecular epidemiology data provided so as to corroborate with discussion as mentioned on the line no 201 to 204.

Response to Reviewer: 

The Molecular epidemiology data has been tabulated in tables 5 & 6, page no: 11 & 12.

• More over its only drug resistance study using different methods including molecular levels and gene detection. 

Response to Reviewer: 

Yes its drug resistance study using different methods including molecular levels and gene detection

• As the study has not been compared with any other groups or with previous year data title of the study is deceptive. Needs to be corrected.

Response to Reviewer: 

Title has been changed included in the Title page.

---

## [Decision Letter · Decision Letter 1]

23 Mar 2020

PONE-D-19-29203R1

Plasmid-mediated antibiotic resistance among uropathogens in primigravid women – Hyderabad, India

PLOS ONE

Dear Dr Kammili,

Thank you for submitting your manuscript to PLOS ONE. After careful consideration, we feel that it has merit but does not fully meet PLOS ONE’s publication criteria as it currently stands. Therefore, we invite you to submit a revised version of the manuscript that addresses the points raised during the review process.

There are some minor comments marked on the text

We would appreciate receiving your revised manuscript by May 07 2020 11:59PM. To enhance the reproducibility of your results, we recommend that if applicable you deposit your laboratory protocols in protocols.io, where a protocol can be assigned its own identifier (DOI) such that it can be cited independently in the future. For instructions see: http://journals.plos.org/plosone/s/submission-guidelines#loc-laboratory-protocols

We look forward to receiving your revised manuscript.

Kind regards,

Iddya Karunasagar

Academic Editor

PLOS ONE

Additional Editor Comments (if provided):

Most of the reviewers comments have been addressed by the authors. There are some minor comments marked on the text

Reviewers' comments:

Reviewer's Responses to Questions

**Comments to the Author**

1. If the authors have adequately addressed your comments raised in a previous round of review and you feel that this manuscript is now acceptable for publication, you may indicate that here to bypass the “Comments to the Author” section, enter your conflict of interest statement in the “Confidential to Editor” section, and submit your "Accept" recommendation.

Reviewer #1: All comments have been addressed

Reviewer #2: (No Response)

2. Is the manuscript technically sound, and do the data support the conclusions?

Reviewer #1: Yes

Reviewer #2: Yes

3. Has the statistical analysis been performed appropriately and rigorously? 

Reviewer #1: Yes

Reviewer #2: I Don't Know

4. Have the authors made all data underlying the findings in their manuscript fully available?

Reviewer #1: Yes

Reviewer #2: Yes

5. Is the manuscript presented in an intelligible fashion and written in standard English?

Reviewer #1: Yes

Reviewer #2: Yes

6. Review Comments to the Author

Reviewer #1: The issues raised by the reviewer has been addressed adequately. The authors have responded to each of the queries raised by the reviewer and made necessary changes in the revised text.

Reviewer #2: though the authors have incorporated replies to most of the queries but some of the queries are not addressed correctly. author needs to modify the same so as to make manuscript more meaningful. see the attchment for details.

7. PLOS authors have the option to publish the peer review history of their article (what does this mean?). If published, this will include your full peer review and any attached files.

Reviewer #1: Yes: Dr. Reba Kanungo MD,PhD Head of Microbiology Pondicherry Institute of Medical Sciences Puducherry-India

Reviewer #2: No

---

## [Author Response · Author response to Decision Letter 1]

19 Apr 2020

1. The beginning of the discussion has been changed to remove the first three lines and begin the discussion with the abstract aims, as suggested. See lines 208-209.

2. In the discussion, we have added specific rates of resistance from other studies in India. See lines 227-229, 246-247, and 256-257.

---

## [Editor Report · Decision Letter 2]

21 Apr 2020

Plasmid-mediated antibiotic resistance among uropathogens in primigravid women – Hyderabad, India

PONE-D-19-29203R2

Dear Dr. Kammili,

We are pleased to inform you that your manuscript has been judged scientifically suitable for publication and will be formally accepted for publication once it complies with all outstanding technical requirements.

With kind regards,

Iddya Karunasagar

Academic Editor

PLOS ONE

Additional Editor Comments (optional):

All reviewer comments addressed satisfactorily
---

## [Editor Report · Acceptance letter]

28 Apr 2020

PONE-D-19-29203R2 

Plasmid-mediated antibiotic resistance among uropathogens in primigravid women – Hyderabad, India 

Dear Dr. Kammili:

I am pleased to inform you that your manuscript has been deemed suitable for publication in PLOS ONE. Congratulations! Your manuscript is now with our production department. 

With kind regards,

on behalf of

Dr. Iddya Karunasagar 

Academic Editor

PLOS ONE